# Recent Trends and Advancements in the Diagnosis and Management of Gastric Cancer

**DOI:** 10.3390/cancers14225615

**Published:** 2022-11-15

**Authors:** Emaan Haque, Abdullah Esmail, Ibrahim Muhsen, Haneen Salah, Maen Abdelrahim

**Affiliations:** 1College of Medicine, Alfaisal University, Riyadh 11533, Saudi Arabia; 2Section of GI Oncology, Houston Methodist Neal Cancer Center, Houston, TX 77030, USA; 3Section of Hematology and Oncology, Department of Medicine, Baylor College of Medicine, Houston, TX 77030, USA; 4Department of Pathology, Houston Methodist Hospital, Houston, TX 77030, USA; 5Cockrell Center for Advanced Therapeutic Phase I Program, Houston Methodist Research Institute, Houston, TX 77030, USA; 6Department of Medicine, Weill Cornell Medical College, New York, NY 10021, USA

**Keywords:** gastric cancer, diagnosis, management, biomarkers, immunotherapy, tyrosine kinase inhibitors, monoclonal antibodies

## Abstract

**Simple Summary:**

Gastric cancer is the fifth most common tumor worldwide. In the past couple of decades, there have been many advancements toward earlier detection and better treatments for this aggressive disease. There are currently many serological tests and biomarkers being investigated to allow for the non-invasive early diagnosis of gastric cancer. The treatment options for this tumor are always rapidly evolving, and we will emphasize the medical treatment options available. We hope to outline and explain some of these latest advancements to allow clinicians and future researchers to have a better understanding of this rapidly changing field and allow them to make informed decisions for the care of their patients.

**Abstract:**

Gastric cancer is an enigmatic malignancy that has recently been shown to be increasing in incidence globally. There has been recent progress in emerging technologies for the diagnosis and treatment of the disease. Improvements in non-invasive diagnostic techniques with serological tests and biomarkers have led to decreased use of invasive procedures such as endoscopy. A multidisciplinary approach is used to treat gastric cancer, with recent significant advancements in systemic therapies used in combination with cytotoxic chemotherapies. New therapeutic targets have been identified and clinical trials are taking place to assess their efficacy and safety. In this review, we provide an overview of the current and emerging treatment strategies and diagnostic techniques for gastric cancer.

## 1. Introduction

As the fifth most common malignant neoplasm and the fourth most common cause of cancer-related death worldwide [1], there is no doubt that gastric cancer (GC) is a disease requiring further insight into how to better diagnose and manage it. Geographically, the incidence of gastric cancer is highest in East Asian and Eastern European regions, with around 60% of all GCs worldwide being seen in East Asia, of which 43.9% are accounted for by China alone [2]. One possible hypothesis for this pattern is that these regions are known to have a high prevalence of the established risk factors for GC, such as a different cytotoxic-associated gene A (*cagA*) strain of *H. pylori* and increased intake of salt-preserved or smoked foods [3]. Other risk factors for GC include smoking and heavy alcohol consumption [2].

Given its high incidence, it is imperative that further research is done to better understand this disease. In this review, we summarize the current status of diagnostic modalities and treatments for gastric cancer.

## 2. Diagnosis

In the United States, about one-third of patients diagnosed with gastric cancer have a distant metastasis at the time of diagnosis [4]. Upper GI series and endoscopy are the gold standard and mainstay in current clinical practice for early gastric cancer diagnosis. In fact, the use of endoscopy for screening is associated with lower gastric cancer-related mortality [5]. However, in the Western world, compared to Korea and Japan where there is a high prevalence of gastric cancer, the use of upper GI endoscopy for screening is cost-ineffective and invasive, and hence other non-invasive, cost-conscious diagnostic methods are being sought [6]. Liquid biopsies have emerged as a non-invasive way of using bodily fluids (i.e., blood, peritoneal lavage, gastric juice/lavage, etc.) to provide early tumor diagnosis, assess prognosis, identify druggable targets, and monitor tumor burden while undergoing treatment [7,8]. There are several advantages to using liquid biopsies as screening for GC, one of the most obvious being its relative non-invasiveness compared with the current gold standard of endoscopy. These blood tests can detect biomarkers which include a variety of molecules that are associated with carcinogenesis of gastric cancer, including proteins, DNA, various types of RNA, exosomes, etc. Figure 1 provides a schematic for some of the liquid biopsy markers currently being studied to diagnose gastric cancer.

### 2.1. Biomarkers

#### 2.1.1. Proteins

Currently, the most widely used biomarkers for the diagnosis and monitoring of gastric cancer are carcinoembryonic antigen (CEA) and carbohydrate antigen (CA) 19-9. Both CEA and CA19-9 have been shown to correlate with tumor burden and depth of tumor invasion [9]. However, both these markers have been proven to have low sensitivity and specificity by many studies and hence have poor diagnostic, prognostic, and monitoring values [9].

Pretreatment CA-125 is another classical prognostic marker for gastric cancer that has been associated with gastric cancer recurrence, rendering it a diagnostic tool to predict poor prognosis. Furthermore, many studies have demonstrated that elevated levels of serum CA-125 correlate with peritoneal dissemination of the primary gastric tumor. Although the sensitivity of this biomarker is poor, ranging from 19.78% [10] in some studies up to 34.3% [11], it has still been shown to have clinical utility, especially in patients with unresectable advanced or recurrent gastric cancer [11].

Another tumor marker that has been gaining significant traction for the purpose of gastric cancer screening and monitoring is CA72-4 [12]. Although there are limited studies assessing its utility, it has been shown in one study to have a higher specificity in indicating the recurrence of gastric cancer compared to CEA and CA19-19 (97% vs. 79% and 74%, respectively) (*p* < 0.05) [12]. However, despite its high specificity, it has been shown to have poor sensitivity and hence has limited clinical value.

Pepsinogen is another protein biomarker that can be used for the screening of gastric cancer. Pepsinogen, a precursor of pepsin, is secreted by the chief cells of the stomach and can be present in two forms, pepsinogen I (PG I), which is secreted from the fundus, and pepsinogen II (PGII), which is secreted from the pylorus and Brunner’s glands of the duodenum [13,14]. In normal conditions, the ratio between PG I and PG II is similar. In atrophic gastritis, which is a risk factor for gastric cancer since it can cause intestinal metaplasia, the destruction of fundic chief cells causes a remarkable reduction in PG I levels and hence causes a low PG I:PG II ratio [15,16]. Hence, a low PG I:PG II ratio is often used to screen for patients with atrophic gastritis which can possibly detect early gastric tumors. However, a major limitation of this biomarker is its poor sensitivity and specificity for detecting gastric tumors, which can be up to 77% and 73%, respectively, in the general population [17]. Furthermore, due to the vast variation in baseline pepsinogen levels based on ethnicity and gender, this practice has yet to be adopted into routine clinical practice [18].

Another promising biomarker is trefoil factor 3 (TFF3). This minute peptide is secreted from the goblet cells of the small and large intestine and can also be secreted from the gastric mucosa that has undergone intestinal metaplasia. Studies have shown high serum TFF3 levels to have a sensitivity of 80.9% and specificity of 81% for gastric cancer [19]. A meta-analysis of 17 studies evaluating the diagnostic value of TTF3 for GC showed that tissue TTF3 expression was associated with a higher risk of lymph node metastasis (OR 2.20, *p* < 0.001), muscularis propria invasion (OR 1.51, *p* = 0.006), and worse TNM stage (OR 2.26, *p* < 0.001) [20]. Furthermore, it has been shown that when combining the measurement of the PG I/PGII ratio with TFF3, there was a higher positive predictive value for detecting GC than when testing for each of the components separately [19,21]. Despite these promising results, currently there are no studies evaluating the utility of TFF3 for gastric cancer screening and diagnosis in clinical practice.

Alpha-fetoprotein (AFP) is another biomarker for gastric cancer that is most commonly seen in a rare subset of AFP-producing gastric carcinomas [22]. AFP is a glycoprotein synthesized from the embryonic yolk sac and liver during pregnancy; in clinical practice, it is most commonly used as a tumor marker for hepatocellular carcinoma [23]. Studies have found that AFP-positive GCs are characterized by a more aggressive behavior than AFP-negative GC tumors, with a higher chance of liver metastasis and venous invasion [23,24]. Hence, it is recommended that physicians routinely check AFP levels in gastric cancer patients, especially if there is concern about liver metastasis [25].

#### 2.1.2. Circulating Tumor Cells

In 1869, a liquid biopsy was performed from the peripheral blood and provided the first known evidence for the presence of circulating tumor cells (CTCs) [26]. Liquid biopsies are samples of blood or other biological fluids that are used to detect and analyze cancer cells or cancer cell-derived molecules [27]. CTCs are cancer cells that have been released either from the primary tumor site or from the metastatic sites [28]. Several meta-analyses have demonstrated an association between the presence of GC CTCs and advanced tumor stage, lymphatic invasion, and poorer survival [29]. There are various markers for gastric cancer CTCs; for the epithelial subtype, they include EpCAm, cytokeratin (CK): CK8, CK18, and CK19 and for the mesenchymal subtype, they include vimentin and twist [30,31]. Another interesting marker for gastric cancer CTCs is fluorescence in situ hybridization (FISH) detection of CTCs with chromosome 8 aneuploidy, a mutation commonly found in gastric tumor cells [32].

Studies have shown that the clinical utility of CTCs is mainly limited to monitoring gastric tumor treatment response and prognosis rather than the early diagnosis of the tumor [28]. For example, the PRODIGE 17 trial conducted in patients with advanced gastric and esophageal cancer demonstrated that the dynamic changes in CTC count between baseline and 4 weeks after treatment were significantly associated with progression-free survival (PFS) and overall survival (OS) [33,34]. However, since CTCs are generally quickly eliminated from the body through the immune system, only a few CTCs survive in the blood circulation (around 1 CTC/mL of blood) and hence this sparsity of CTCs leads to challenges in accurately detecting their presence from liquid biopsies [35]. Due to this phenomenon as well as the heterogeneous nature of CTCs, various CTC detection methods have yielded different detection rates.

#### 2.1.3. Circulating Tumor DNA

Circulating tumor DNA (ctDNA) is another biomarker that can be extracted from liquid biopsies for the diagnosis and monitoring of gastric cancer. CtDNA can be produced from primary tumor cells, CTCs, or a distant metastasis and it can give a wide range of information on the malignancy, such as methylation status and other genetic alterations [28]. Fang et al. demonstrated a correlation in gastric cancer patients between ctDNA levels and vascular invasion, 5-year survival rate, and peritoneal recurrence [36]. Furthermore, a meta-analysis of 16 studies showed a significant association between the presence of ctDNA with worse OS (*p* < 0.001) and DFS (*p* < 0.001) [37]. The study also showed that ctDNA levels had a significant association with TNM stage, tumor depth, lymph node metastasis, and distant metastasis with a specificity of 95% and a sensitivity of 62%. However, due to the complex technology required to detect ctDNA within the plasma, it has yet to be of significant use in clinical practice [28].

#### 2.1.4. Non-Coding RNA

Non-coding RNAs (ncRNAs) are types of RNA that do not encode protein and can be classified into two subgroups: small ncRNAs (sncRNAs) and long ncRNAs (lncRNAs). The sncRNAs can be further subclassified into microRNAs (miRNAs), small nuclear RNAs (snRNAs), and piwi-interacting RNAs (piRNAs). These various non-coding RNAs can also be used in the detection and monitoring of gastric cancer.

miRNAs play a crucial role in various cellular functions through regulating epigenetic mechanisms; these functions include cellular growth, apoptosis, differentiation, and even gastric tumor carcinogenesis. Wu et al. studied 50 GC patients and 50 patients without GC and found an increase in levels of miRNA-21 in patients with GC compared to those who did not have it (*p* < 0.01), with a sensitivity of 81.3% and a specificity of 73.4% [38]. Hung et al. demonstrated increased levels of miRNA-376c in the tissue, plasma, and urine of GC patients owing to the fact that miRNA-37c was found to increase the proliferation, migration, and anchorage-independent growth of carcinoma cells [39]. Other panels of miRNA that have also been shown to be upregulated include miRNA-196a [40], -200c [41], -375 [42], -940 [43], and many others [28]. Despite these promising results, the clinical utility of miRNA in routine practice has many current limitations. For example, there can be inaccuracies in miRNA quantification due to variations in processing, storage, RNA extraction, and reference gene choice during qRT-PCR since there is no unique protocol developed yet to control these parameters [32]. More details about the utility of miRNA in the diagnosis of GC are outlined in the next section on circulating extracellular vesicles.

Another sncRNA that is used as a biomarker in the diagnosis of gastric cancer is piRNAs. This newly discovered type of non-coding RNA has been shown to be a molecule that is not easily degraded and able to be detected in various human bodily fluids, including serum and gastric juice [44]. Cui et al. [45] performed a peripheral blood test in both healthy and GC patients and showed that GC patients had lower levels of piRNA-651 and piRNA-823 compared to their healthy counterparts. These results also had relatively high sensitivity and specificity of 94.9% and 96.4%, respectively. Due to these promising results, further studies and clinical trials are being conducted to better understand the clinical utility of piRNAs as potential biomarkers for gastric cancer [46].

LncRNAs have also been proposed as biomarkers for GC. For example, a lncRNA called “high up-regulated in liver cancer” (HULC) has been shown to be increased in the serum of GC patients compared to normal controls [47]. Likewise, the lncRNA H19 also showed similar results [48]. Supporting this evidence, both serum HULC and H19 were shown to be significantly decreased in post-treatment GC patients compared to levels obtained prior to treatment. The sensitivity for HULC and H19 was 82% and 74%, respectively, while the specificity for both molecules was 83.6% and 58%, respectively. The clinical application limitations are similar to those of miRNA.

#### 2.1.5. Circulating Extracellular Vesicles

Extracellular vesicles (EVs), also known as exosomes, are small spherical structures with an outer lipid bilayer that are secreted from cells into the extracellular space, and participate in inter-cellular communication through the transfer of functional molecules scavenged and secreted into EVs [8,49]. Exosomes are a type of EV measuring 40–120 nm that are produced in the endosomal compartment of the cell [50]. The contents of these exosomes include proteins, miRNAs, lncRNAs, etc. GC-derived exosomes can communicate with cells in the tumor microenvironment, allowing it to become more favorable in establishing metastatic niches. These exosomes also suppress host innate and adaptive immune responses by regulating host immunomodulatory mediators [8,51].

Some exosomal proteins are involved in the development of GC. TGF-β1 is an immunosuppressive cytokine that has been detected in exosomes of GC patients and was found to be correlated with lymphatic metastasis [52,53]. Tripartite motif 3 (TRIM3) is a protein that normally inhibits the proliferation of GC cells; it has been found that the levels of TRIM3 in serum exosomes of patients with GC are lower than those of healthy controls, making it a potential diagnostic biomarker for GC [54]. Gastrokine-1 (GKN-1) is another exosomal cargo protein involved in regulating the immune response and inhibiting proliferation of GC cells [54]. Yoon et al. found that healthy controls had significantly higher serum GKN1 levels than GC patients (6.34 ng/μL vs. 3.48 ng/μL, *p* < 0.0001), suggesting it to be another potential biomarker for GC. Heat shock proteins (HSP) 60 and 70 have been found in higher concentration within exosomes derived from malignant ascites in GC patients compared with exosomes derived from ascites in non-GC patients [55]. HSP-60 and 70 aid in the immune response against GC by promoting the maturation of dendritic cells, inducing a cytotoxic T-lymphocyte response against the tumor [54]. These exosomal proteins have yet to be studied in large cohort clinical trials and hence their applicability in the clinical setting is yet to be known.

The use of exosomal DNA for the diagnosis and prognosis of GC is an area of research rarely targeted in the literature. There have been only three studies investigating this up until now with only four exosomal genes identified in relation with GC so far: *BARHL2*, *LINE1*, *SOX17*, and miRNA-34b/c gene. Gastric juice-derived exosomal *BARHL2* gene methylation was suggested to have promising potential as a biomarker with GC patients being more likely to have *BARHL2* methylation compared to non-GC controls (90% sensitivity and 100% specificity) [56]. Another study also investigating the detection of methylated DNA in gastric juice-derived exosomes found that patients with GC had reduced *LINE1* methylation whereas *SOX17* gene methylation was detected in both early and advanced gastric cancer of both intestinal and diffuse type [57]. These findings suggest the promising potential of gastric juice-derived exosomal DNA for the early detection of GC in the clinical setting.

It has been proposed that exosomal miRNAs have promising potential as diagnostic molecules for GC tumors. Ren et al. extracted exosomes from GC cell lines and non-GC cell lines and found that the exosomes of GC cell lines contained higher levels of miRNA-21-5p and miRNA-30-p compared to the non-GC cell lines [58]. Another study by Wang et al. found that exosomal miRNA-19b-3p and exosomal miRNA-106a-5p had 95% sensitivity and 90% specificity in detecting GC, suggesting them to be promising biomarkers for the diagnosis of GC [59]. Huang et al. identified six miRNAs that were significantly upregulated in the serum of GC patients, with four of them (miRNA-10b-5p, miRNA-195-5p, miRNA-20a-3p, and miRNA-296-5p) showing significant upregulation in serum exosomes [60]. Furthermore, Tokuhisa et al. found that miRNA-1225-5p and miRNA-21 from peritoneal lavage fluid were upregulated in the later stages of GC and correlated with serosal invasion, which could potentially predict peritoneal recurrence following curative GC resection [61]. Despite these extensive findings on the potential utility of exosomal miRNAs for the diagnosis and prognosis of GC, there have yet to be any clinical trials to investigate this further in the clinical setting and hence their applicability in the real word is yet to be determined [51,62].

## 3. Treatment

Although there is a wide range of therapies available for the management of gastric cancer, the molecular and clinical heterogeneity associated with the disease has led to newer classifications of GC patients which provide therapeutic approaches based on the genome and clinical evidence. Current guidelines recommend all patients eligible for systemic treatment undergo molecular profiling to determine the appropriate therapy and treatment strategy.

### 3.1. Epidermal Growth Factor Receptors

The human epidermal growth factor receptor (ErbB or EGFR) family is composed of four types of tyrosine kinase receptors (TKRs): EGFR (ErbB-1 or HER-1), HER-2 (ErbB-2), HER-3 (ErbB-3), and HER-4 (ErbB-4). These receptors play a critical role in cell growth, proliferation, and migration of tumors [63]. It has been known that gastric tumors express HER in a heterogeneous pattern, especially with HER-1 and HER-2. HER-1 is amplified in 27–64% of gastric tumors [64,65] whereas HER-2 is amplified in 30% of tumors [66]. Table 1 and Table 2 list phase II and phase III clinical trials that studied the effects of targeting the HER-1 and HER-2 receptors in gastric cancer patients.

#### 3.1.1. HER-1

Normally, when a ligand (i.e., EGF, TGFα, amphiregulin, epiregulin, etc.) binds EGFR, it induces tyrosine phosphorylation, which stimulates multiple downstream signaling cascades that in turn promote cell proliferation, angiogenesis, migration, survival, and adhesion. Deregulation of EGFR signaling can occur through multiple mechanisms such as receptor overexpression, activating mutations, and gene copy numbers (GCNs) [92].

Although EGFR amplification has been shown to occur in a significant proportion of gastric cancers, there is no general consensus on its prognostic value. Some studies suggest that a higher overexpression is associated with poorer outcomes [93,94] while others suggest the complete opposite [95]. Since EGFR is a well-recognized mediator for the oncogenic phenotype of gastric cancer [96], many EGFR targeting agents have entered clinical practice, albeit with disappointing results. The first category of anti-EGFR therapeutic agents are tyrosine kinase inhibitors (TKIs) which have greater efficacy in tumors with activating EGFR mutations such as non-small cell lung cancer. The other category of anti-EGFR therapeutic agents are monoclonal antibodies which are effective in tumors that overexpress EGFR, regardless of whether the EGFR is actually mutated or not [92]. Anti-EGFR monoclonal antibodies have multiple mechanisms to induce anti-tumor activity, such as antibody-dependent cell-mediated cytotoxicity (ADCC), competitive inhibition of ligand binding, receptor endocytosis/internalization/degradation, and complement-mediated cytotoxicity [97].

Early phase II clinical trials have suggested a potential benefit for the use of EGFR inhibitors in patients with gastric cancer. For example, Richards et al. [68] demonstrated that the addition of cetuximab to combination chemotherapy of doxetaxel + oxaliplatin as a second-line therapy for the management of metastatic gastric cancer resulted in a higher mPFS of 5.1 months in the cetuximab therapy group, compared to 4.7 months in the combination chemotherapy alone group (*p* > 0.05). However, two larger phase III randomized trials, REAL3 and EXPAND, demonstrated that there was no improvement in survival for patients with advanced gastric cancer treated with anti-EGFR therapy [74,75]. In fact, the REAL3 trial showed a statistically significant (*p* = 0.013) worse mOS in the panitumumab subgroup and the EXPAND trial showed a statistically significant (*p* = 0.032) worse mPFS in the cetuximab subgroup. Hence the evaluation of EGFR inhibition was abruptly abandoned for gastric cancer.

However, one caveat to the phase III trials was that there was no patient selection performed on the basis of EGFR amplification/overexpression which rendered the results as questionable due to the heterogeneity in the expression of EGFR in gastric cancer [98]. To address this, Smyth et al. [99] tested EGFR copy numbers in tissue and liquid biopsies taken from the patients evaluated in the REAL3 trial. The results showed that only 7% of patients in the trial were EGFR-amplified and that the use of anti-EGFR therapy (panitumumab) in these EGFR-amplified patients actually worsened the prognosis (although not statistically significant, likely due to the small number of EGFR-amplified cases). Furthermore, the data showed an antagonistic effect when anthracycline chemotherapy was combined with anti-EGFR therapy. These relatively consistent overall findings suggest that EGFR inhibition probably does not represent an important therapeutic target for most patients with advanced gastric cancer.

#### 3.1.2. HER-2

HER-2 is a proto-oncogene that encodes the transmembrane receptor-like HER2 protein. When activated, it initiates signaling pathways that lead to cell proliferation, differentiation, and vascular and lymphatic angiogenesis [100]. HER-2 overexpression is determined through immunohistochemistry (IHC) and/or fluorescence in situ hybridization (FISH). The IHC score has three categories depending on the degree of HER-2 amplification: negative (0+ or 1+), equivocal (2+), or positive (3+). Amplification of this proto-oncogene has been associated with poor prognosis and constitutes a predictive factor for poor response to chemotherapy [101]. Hence, targeting HER-2 in HER-2-positive gastric cancer is a plausible therapeutic approach. Similar to therapeutic agents targeting HER-1, the drugs targeting HER-2 can also be categorized as either anti-HER-2 monoclinal antibodies and HER-2 targeting TKIs.

The anti-HER-2 monoclonal antibody, Trastuzumab, has been the only well-established cornerstone management for many years for advanced HER-2 positive gastric cancer. The landmark phase III ToGA trial [82] conducted in 2010 compared the efficacy of trastuzumab in combination with the standard first-line chemotherapy regimen at the time (cisplatin + 5-FU) versus chemotherapy alone, and the trastuzumab with chemotherapy combination was shown to have a statistically significant improved OS, PFS, and ORR compared to chemotherapy alone. However, one limitation of the ToGA study was the fact that about one-third of the patients assigned to the trastuzumab arm were underdosed, which was theorized to have caused a worse survival. Hence, the phase III HELOISE trial [83] was conducted to assess whether there was any difference in efficacy when chemotherapy was combined with low-dose trastuzumab compared to high-dose trastuzumab. However, the authors found that the high-dose regimen did not result in improved OS or PFS. Following the ToGA study, another phase II trial studied the efficacy of combining trastuzumab with other chemotherapy regimens, such as replacing cisplatin with oxaliplatin and 5-FU with capecitabine. These results [76,77,78,79,80] showed similar results in terms of efficacy as the ToGA trial, and hence these chemotherapy combinations are also used along with trastuzumab for the first-line treatment of HER-2-positive advanced gastric cancer.

Another monoclonal antibody investigated for use in AGC is pertuzumab. It has a similar mechanism of action as trastuzumab except that pertuzumab binds to the dimerization domain of HER-2, which prevents HER-2 heterodimerization with other HER family receptors, whereas trastuzumab binds to the transmembrane domain, which prevents HER-2 dimerization [102]. The phase III JACOB trial evaluated the addition of pertuzumab vs. placebo to trastuzumab with chemotherapy in the first-line setting. However, although the mPFS and ORR showed statistically significant improvement, there was no statistically significant improvement in mOS, which was the primary endpoint [84]. These findings highlight the heterogeneity in HER-2 biology in gastric cancer vs. breast cancer and hence the varying efficacy of targeted therapy in both tumors.

Antibody–drug conjugates (ADC) are another therapeutic strategy currently being investigated for AGC. Trastuzumab emtansine (T-DM1) is an ADC consisting of the anti-HER-2 monoclonal antibody trastuzumab with the tubulin inhibitor emtansine. Emtansine is released into HER-2-positive tumor cells to cause mitotic arrest and apoptosis [100]. However, the phase III GATSBY trial demonstrated no statistically significant survival benefit of T-DM1 compared to standard taxane therapy [87]. Another ADC proposed for the management of AGC is trastuzumab deruxtecan (T-DXd) which combines trastuzumab with deruxtecan, a topoisomerase I inhibitor that when entering tumor cells leads to the inhibition of DNA replication resulting in cell cycle arrest and tumor cell apoptosis [103]. A phase II DESTINY trial [85] showed that T-DXd had a statistically significant improved OS (12.5 months vs. 8.4 months, *p* = 0.01) and ORR (51% vs. 14%, *p* < 0.001) compared to patients on chemotherapy alone, in patients with AGC as a third-line or later therapy. Following this study, the FDA approved T-DXd for use in AGC after failure with a trastuzumab-containing regimen. RC48 is another ADC that linked humanized anti-HER-2 IgG1, a valine–citrulline linker, and MMAE (a microtubule inhibitor) together, and a phase II study [86] found an ORR of 18.1% (95% CI: 11.8–25.9%) and a mOS of 7.6 months (95% CI: 6.6–9.2) in patients with HER-2 overexpressing AGC. RC48 is still currently being investigated in further clinical trials.

TKIs are another category of drugs used to target HER-2 in AGC. Lapatinib is a small molecule TKI that inhibits both EGFR and HER-2, which results in reduced intracellular signaling and hence suppressed tumor proliferation [104]. The phase III LOGiC trial examined the use of lapatinib as a first-line treatment when combined with CAPEOX compared to CAPEOX alone and found no statistically significant improvement in the primary endpoint, which was mOS (12.2 months vs. 10.5 months, *p* = 0.91), although the PFS (6.0 months vs. 5.4 months, *p* = 0.038) and ORR (53% vs. 39%, *p* = 0.0031) were significant [90]. The TyTAN trial examined the efficacy of paclitaxel with or without lapatinib in the second-line setting; however, there was no statistically significant improvement in mOS (11.0 months vs. 8.9 months, *p* = 0.1044) or mPFS (5.4 months vs. 4.4 months, *p* = 0.2441), despite a significant ORR (27% vs. 9%, *p* < 0.001) [91]. Dacomitinib is another TKI that had a phase II trial conducted, yet the results showed that there was no substantial therapeutic benefit for its use in HER-2-positive AGC [88].

### 3.2. Angiogenesis

The process of angiogenesis is modulated by the interaction of VEGF with its TKRs, known as VEGFRs. There are four types of VEGF (VEGF-A, VEGF-B, VEGF-C, and VEGF-D) that have been identified along with three types of VEGFRs (VEGFR-1, VEGFR-2, and VEGFR-3) [105].

Therapies targeting this pathway can be either monoclonal antibodies or TKIs. Even though there is no measurable predictive factor to determine which patients respond better to VEGF pathway inhibition, expression of VEGF has been seen to occur in almost 48% of gastric cancers and is associated with poorer prognosis [106]. Several clinical trials have shown that there is clinical benefit when targeting the VEGF/VEGFR pathway. Table 3 outlines the results of phase II and III trials against this molecular target.

Bevacizumab is an anti-VEGF-A monoclonal antibody that inhibits circulating VEGF-A activity [146]. Although several phase II trials [107,108,109,110,111,112,113,114,115,116,117] have suggested that the combination of Bevacizumab with chemotherapy could possibly provide some clinical benefit for AGC patients, the phase III trials “AVAGAST” [121] and “AVATAR” [122] concluded that there was no significant difference in mOS between patients taking chemotherapy alone and patients taking chemotherapy along with bevacizumab. Interestingly, a sub-analysis of the AVAGAST trial [147] found that non-Asian patients who received bevacizumab in combination with chemotherapy had better outcomes than Asian patients. Ramucirumab is another human monoclonal antibody used for the management of AGC that works by blocking VEGFR-2 [148]. The phase III studies “REGARD” [124] and “RAINBOW” [125] demonstrated statistically significant improvements when ramucirumab was used as a second-line therapy, either alone or in combination with paclitaxel, respectively. However, there was no statistically significant difference seen in mOS when ramucirumab was combined with chemotherapy as a first-line therapy, as seen in the “RAINFALL” [123] trial.

Many phase II trials [126,127,128,129,130,131,132,133,134,135,136,137,138,139,140] investigating the efficacy of VEGFR TKIs have shown a lack of survival benefit for AGC. However, the phase II trial investigating apatinib [136] showed promising results when used in the 850 mg dose, one daily in the third-line and beyond settings. This prompted a phase III study which showed statistically significant improvements in mOS and mPFS compared to placebo [144]. In 2014 and 2017, the Chinese and US FDA approved the use of apatinib for the treatment of AGC [149].

### 3.3. Immune Checkpoint Inhibitors

Evasion of the immune system is an established hallmark of cancer [150]. Programmed cell death protein-1 (PD-1) and cytotoxic T lymphocyte protein 4 (CTLA-4) are inhibitory pathways critical for maintaining self-tolerance. When PD-1, a negative co-stimulatory receptor expressed on activated T-cell surfaces, binds to its ligands, programmed cell death ligands 1 and 2 (PD-L1/L2), leading to an inhibition of cytotoxic T-cell response, which allows tumor cells to escape T-cell-induced anti-tumor activity. CTLA-4, another receptor found on T-cells, binds to B7 on antigen-presenting cell surfaces, which prevents the B7 from biding with the co-stimulatory CD28 receptor, preventing T-cell activation.

In the “era of revolution” in cancer management with immunotherapy, there have been attempts to integrate immune checkpoint inhibitors in the therapeutic algorithm for AGC. Gastric tumors that are Epstein–Barr virus (EBV) positive and microsatellite-unstable (MSI) have been shown to be potentially most responsive to immunotherapy drugs [151]. EBV-positive GC (represents up to 9% of all GC tumors) is associated with programmed death ligand 1 (PD-L1 gene amplification, which suggests higher immunogenicity and hence is more likely to respond to immune checkpoint inhibition. MSI tumors (which represent up to 15–30% of all GC tumors) are characterized by a lymphocytic infiltrate which may reflect the activation of T-cells against tumor antigens and genomic changes in tumor cells linked to PD-L1 expression, hence indicating a potential role for immunotherapy [152,153]. Furthermore, both EBV and MSI-positive GC tumors have a high somatic mutational burden which is also a feature associated with response to immunotherapy. Table 4 outlines the results of phase II and III trials using immune checkpoint inhibitors in gastric cancer.

Blocking the PD-1/PD-L1 interaction can enhance the immune response against tumors. Pembrolizumab is a humanized IgG4 monoclonal anti-PD-1 antibody. The phase II KEYNOTE-059 [154] trial showed clinical benefit when using pembrolizumab monotherapy in the second-line setting and beyond for AGC. This led to it becoming FDA-approved in 2017 as a third-line treatment for patients with a PD-L1 combined positive score (CPS) ≥ 1 AGC. However, the phase III KEYNOTE-061 [164] and KEYNOTE-062 [165] trials demonstrated that pembrolizumab was non-inferior to chemotherapy, both when used as a monotherapy and in combination with chemotherapy drugs, in the second- and first-line setting, respectively. In the phase III ATTRACTION-2 study [167], Nivolumab, another anti-PD-1 monoclonal antibody, was tested in Asian patients as a monotherapy in the second-line and beyond setting and showed statistically significant improvements in mOS and mPFS, which led to its approval in Japan as a third-line treatment for gastric cancer. Furthermore, the phase III trial CheckMate-649 [168] also showed significant improvements in mOS and mPFS when nivolumab was combined with standard first-line chemotherapy compared to the use of chemotherapy alone. Several other phase II trials have been conducted for other PD-1 inhibitors, such as camrelizumab [160], sintilimab [161], toripalimab [162], and tislelizumab [163]; however, none so far have produced results warranting further phase III trials.

Avelumab is an anti-PD-L1 monoclonal antibody that was investigated in the phase III JAVELIN Gastric 100 study [170] as a maintenance treatment after the first-line chemotherapy in AGC patients; however, it failed to show any significant improvement in mOS or mPFS. Ipilimumab is an anti-CTLA-4 antibody that has been shown to cause a statistically significant worse mPFS when combined with chemotherapy, compared to the use of chemotherapy alone in AGC patients [171].

Immune checkpoint inhibitors have also been combined with other targeted therapies to produce promising results. For AGC, studies have mainly investigated the combination of antu-HER-2 monoclonal antibodies and VEGF/VEGFR inhibitors with immunotherapy. The phase II PANTHERA [155] study investigated the use of pembrolizumab combined with trastuzumab and chemotherapy to treat HER-2 AGC patients in the first-line setting and showed an ORR of 76.7%, with 56.6% of patients showing a reduction in over 50% of the tumor burden. These findings concurred with the ones seen in another phase II study assessing the efficacy of combining trastuzumab with pembrolizumab in AGC patients [156]. These promising results prompted the phase III KEYNOTE-811 [166] study assessing the use of pembrolizumab in the first-line setting combined with trastuzumab and chemotherapy, and showed to have an ORR of 74.4% in the intervention arm vs. 51.9% in the arm with trastuzumab and chemotherapy only (*p* = 0.00006). The results of the KEYNOTE-811 study led the FDA to grant accelerated approval on pembrolizumab plus trastuzumab and chemotherapy for first-line treatment of HER-2-positive gastric cancer in May 2021 [175].

Margetuximab is a novel monoclonal antibody that binds to the same HER-2 dimerization domain as trastuzumab but, in contrast to trastuzumab, it has increased binding to the activating Fcy receptor IIIa and decreased binding to the Fcy receptor IIb, which results in enhanced anti-tumor activity compared to trastuzumab [176]. Furthermore, margetuximab has also been shown in in vitro studies to upregulate PD-L1 expression in tumor cells. This unique mechanism of action resulted in investigators conducting a phase Ib/II trial to assess the efficacy of combining margetuximab with pembrolizumab in HER-2-positive AGC [81], which showed promising results with an ORR of 18.5% and a DCR of 53%, in turn prompting investigators to conduct a phase II/III “MAHOGANY” trial [177] that is still ongoing.

NivoRAM was a phase I/II study that investigated the efficacy of combining nivolumab with paclitaxel and ramucirumab as a second-line treatment of AGC and the results showed patients to have an ORR of 26.7% [159]. The EPOC1706 study [142] was a phase II trial that examined the efficacy of combining lenvatinib with pembrolizumab in the first-line setting and beyond for AGC and the results showed promising results, with patients having an ORR of 69%, especially patients who had high PD-L1 expression (CPS ≥ 1 subgroup: 84%; CPS ≥ 10 subgroup 100%). The LEAP-005 phase II trial [143] which also studied the efficacy of combining lenvatinib with pembrolizumab concurred with the results of EPOC1706.

Combinations of different immune checkpoint inhibitor medications, mainly PD-1/PD-L1 inhibitors with CTLA-4 inhibitors, have also been explored for gastric cancer. The rationale behind this combination could be due to the fact that one of the causes of resistance to PD-1/PD-L1 blockade is the presence of immune suppression through other immune checkpoints, such as CTLA-4, which is a key negative regulator of anti-tumor T-cell response [178]. The phase I/II CheckMate-032 study [172] randomized patients to nivolumab monotherapy (3 mg/kg) and nivolumab with ipilimumab (in two different doses—1 mg/kg nivolumab + 3 mg/kg ipilimumab or 3 mg/kg nivolumab + 1 mg/kg ipilimumab) and found a higher ORR in the combination nivolumab (1 mg/kg) with ipilimumab (3 mg/kg) group compared to the other subgroups, hence supporting the hypothesis that the addition of CTLA-4 inhibitors could improve response to PD-1/PD-L1 inhibitors. Additionally, the promising phase Ib/II “AK104” study [173] examining the efficacy of carmelilumab (a combined PD-1/CTLA-4 inhibitor drug) used in combination with first-line chemotherapy for AGC patients showed inspiring results, with an ORR of 65.9% (2.3% complete, 63.6% partial), disease control rate of 92.0%, mPFS of 7.10 months, and mOS of 17.4 months. This exciting study prompted another phase III study, which is still ongoing, to further examine these findings. In contrast to these studies, another phase Ib/II trial [174] examined the use of durvalumab (PD-L1 inhibitor) with tremelimumab (CTLA-4 inhibitor) combined and as monotherapies, but found no significant response rates in any subgroups.

### 3.4. Anti-DNA Synthesis

TAS-102 is an oral cytotoxic drug composed of trifluridine (TFD), an analog of the thymidine-based nucleoside which inhibits tumor cell growth by being incorporated into DNA during DNA synthesis, and tipiracil (TPI), a molecule which inhibits the metabolism of TFD, thereby prolonging its ability to exert effect [179]. Table 5 outlines the results of phase II and III trials against this molecular target. A phase III trial “TAGS” [180] demonstrated an impressively prolonged mOS in the TAS-102 subgroup vs. placebo in AGC patients in the second-line and beyond setting. Furthermore, a subgroup analysis of the TAGS study investigated the efficacy of this treatment in patients using it as a third-line and fourth-line treatment and found statistically significant improvements in mOS and mPFS compared to placebo [181]. These promising results led to the approval of TAS-102 (trifluridine/tipiracil) as a third-line treatment option in AGC [182]. A recently published phase II study [179] also investigated the combination of TAS-102 with ramucirumab and found modest activity in AGC patients, requiring further investigation.

### 3.5. Anti-Hepatocyte Growth Factor Receptor (Anti-HGFR)

The mesenchymal-epithelial transition factor receptor (c-MET) is a proto-oncogenic receptor tyrosine kinase that is activated by hepatocyte growth factor (HGF). Activation of c-MET receptor promotes tumor formation through increased mitosis and inhibition of apoptosis. C-MET overexpression and gene amplification is a marker of poor prognosis in gastric cancer [184]. Rilotumumab is a humanized IgG2 monoclonal antibody that targets HGF, hence blocking the binding of HGF to c-MET [185]. Table 6 outlines the results of phase II and III trials against this molecular target.

Although phase II studies [186] have suggested a possible benefit with the addition of rilotumumab to first-line chemotherapy, the phase III trial “RILOMET-1” [190] showed a statistically significant worse mOS, mPFS, and ORR in the group taking rilotumumab. The study was ultimately terminated early due to the increased number of deaths due to complications in patients treated with rilotumumab compared to placebo. Onartuzumab is a recombinant humanized anti-c-Met monoclonal antibody [185]. However, phase III trials [191] failed to show any clinical benefit with its addition to first-line chemotherapy. Emertuzumab is a humanized IgG4 monoclonal anti-Met antibody that prevents HGF from binding to c-Met and also degrades c-MET [185]. Phase II trials [188,189] have demonstrated that it may have some anti-tumor activity, although further studies are needed to investigate this.

Some tyrosine kinase inhibitors of the c-Met/HGF pathway, such as foretinib [192] and tivantinib [193], have also been studied in phase II trials; however, none have produced any significant clinical benefit warranting further studies.

### 3.6. Anti-FGFR

The fibroblast growth factor receptor (FGFR) has four family members: FGFR-1, FGFR-2, FGFR-3, and FGFR-4. Of these, FGFR-2 has been shown to be the most frequently amplified and altered in gastric cancer, being overexpressed in around 2–30% of GCs [194]. Table 7 outlines the results of phase II and III trials against this molecular target. The FGFR1-2-3 TKI termed “AZD4547” was investigated in the phase II “SHINE” trial and failed to show any improvement in clinical outcomes. Following this, the phase II “FIGHT” trial [195] investigated the use of the anti-FGFR2b monoclonal antibody, bemarituzumab, in combination with first-line chemotherapy and found promising results with a statistically significant improvement in mPFS. The results of this study prompted a phase III trial for AGC which is still ongoing.

### 3.7. PARP Inhibitors

Poly (ADP-ribose) polymerase (PARP) is important for DNA single-strand break repairs. In tumor cells that possess homologous recombination deficiency, inhibition of PARP can lead to the formation of single-strand breaks which are then transformed into DNA double-strand breaks (that are unable to be repaired through homologous recombination), ultimately leading to genomic instability and tumor cell death [197]. Table 8 outlines the results of phase II and III trials against this molecular target. Despite promising results in phase II trials [198], the phase III trials [199] investigating the use of olaparib with paclitaxel compared with paclitaxel failed to show any statistically significant clinical benefit.

### 3.8. Anti-MMP-9

Matrix metalloproteinase-9 (MMP-9) is known to promote wound healing through collagen deposition as well as activation of cytokines and growth factors. MMP-9-mediated cleavage of cytokines such as interleukin (IL)-8 and IL-1β can induce tumor growth. MMP-9 also cleaves and activates growth factors such as VEGF and FGF-2. Inhibition of MMP-9 can suppress the tumor micro-environment and reduce tumor growth [201]. However, the phase III trial “GAMMA-1” [202] failed to show any statistically significant clinical benefit. A phase II trial [201] was also conducted to demonstrate the efficacy of PD-1 inhibitor nivolumab with andecaliximab compared to andecaliximab alone and also failed to show any clinical benefit with the addition of andecaliximab. Table 9 outlines the results of both of these phase II and III trials against MMP-9.

### 3.9. mTOR Inhibitors

The phosphatidylinositol 3-kinase (PI3K)/Akt and mammalian target of rapamycin (mTOR) is activated in 30% and 60% of GCs, respectively, and is associated with tumor progression [202]. Everolimus is an oral mTOR inhibitor that was evaluated for its efficacy in AGC, however, phase III clinical trials [203] failed to show any clinical benefit. Table 10 outlines the results of both of these trials.

## 4. Conclusions

In conclusion, there have been significant recent developments in the detection and treatment of GC. Although there has been improvement, there are still numerous challenges. The amount of clinical data is growing every day but despite this, there are currently not enough high-quality, well-designed multi-center prospective trials available. Furthermore, the enormous inter-tumor and intra-tumor heterogeneity of GC across individuals and populations results in a lag in transitioning the current molecular research into clinical practice for patient benefit [205]. The diagnosis and treatment approaches used in the East and West also differ significantly [206]. The inconsistency between the approaches used globally limits the advancements toward earlier diagnosis and more effective therapy. Hence, in the future, more cross-disciplinary and international collaboration is needed.

## Figures and Tables

**Figure 1 cancers-14-05615-f001:**
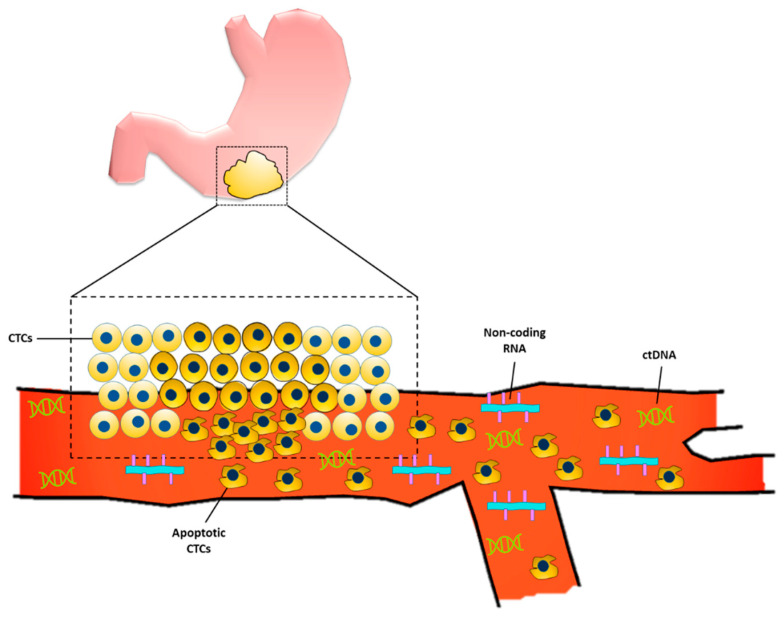
Liquid biopsy markers for gastric cancer. Primary gastric tumor sheds circulating tumor cells (CTCs) into the bloodstream. Some of the CTCs undergo apoptosis which allows for the release of the cell’s genetic material, including circulating tumor DNA (ctDNA) and non-coding RNAs.

**Table 1 cancers-14-05615-t001:** Phase II and III trials of therapies targeting human epidermal receptor-1 (HER-1). mOS, median overall survival; mPFS, median progression-free survival; ORR, objective response rate; M, months.

Drug	Trial	Phase	Method	mOS (*p*-Value)	mPFS (*p*-Value)	ORR (*p*-Value)
Matuzumab	MATRIX EG (first-line) [67]	II	Chemotherapy with or without matazumub	9.4 M vs. 12.2 M (*p* = 0.945)	4.8 M vs. 7.1 M (*p* = 0.678)	31.0% vs. 58.0% (*p* = 0.994)
Cetuximab	DOCOX + C/second-line [68]	II	Chemotherapy (docetaxel + oxaliplatin) with or without cetuximab	9.4 M vs. 8.5 M (*p* > 0.050)	5.1 M vs. 4.7 M (*p* > 0.050)	38.0% vs. 26.0% (*p* > 0.050)
Panitumumab	ATTAX3/second-line [69]	II	Chemotherapy with or without panitumumab	10.0 M vs. 11.7 M (*p* > 0.050)	6.0 M vs. 6.9 M (*p* > 0.050)	57.9% vs. 48.7% (*p* > 0.050)
Panitumumab	first-line [70]	II	Single-arm: Chemotherapy with panitumumab	11.3 M	6.9 M	35.0%
Panitumumab	MEGA/first-line [71]	II	Chemotherapy with or without either panitumumab or rilotumumab	8.3 M (P) vs. 11.5 M (R) vs. 13.1 M (C)	4-month PFS rate:57% (P) vs. 61% (R) vs. 71% (C)	43% (P) vs. 49% (R) vs. 52% (C)
Nimotuzumab	NCS/first-line [72]	II	Chemotherapy with or without nimotuzumab	10.2 M vs. 14.3 M (*p* = 0.062)	4.8 M vs. 7.2 M (*p* = 0.011)	54.8% vs. 58.1% (*p* = 0.798)
Erlotinib	first-line [73]	II	Single arm: Chemotherapy with erlotinib	11.0 M	5.5 M	51.5%
Panitumumab	REAL3/first-line [74]	III	Chemotherapy with or without Panitumumab	8.8 M vs. 11.3 M (*p* = 0.013)	6.0 M vs. 7.4 M (*p* = 0.068)	46% vs. 42% (*p* = 0.42)
Cetuximab	EXPAND/first-line [75]	III	Chemotherapy with or without cetuximab	9.4 M vs. 10.7 M (*p* = 0.95)	4.4 M vs. 5.6 M (*p* = 0.032)	30% vs. 29% (*p* = 0.77)

**Table 2 cancers-14-05615-t002:** Phase II and III trials of therapies targeting human epidermal receptor-2 (HER-2). mOS, median overall survival; mPFS, median progression-free survival; ORR, objective response rate; M, months; ADC, antibody–drug conjugate; TKI, tyrosine kinase inhibitor; CapeOx, capecitabine and oxaliplatin.

Drug Class	Drug	Trial	Phase	Method	mOS (*p*-Value)	mPFS (*p*-Value)	ORR (*p*-Value)
Monoclonal antibody	Trastuzumab	WJOG7112G (T-ACT)/second-line [76]	II	Chemotherapy (paclitaxel) with or without trastuzumab	10.2 M vs. 9.95 M (*p* = 0.20)	3.68 M vs. 3.19 M (*p* = 0.33)	33.3% vs. 31.6% (*p* = 1.00)
	Trastuzumab	first-line [77]	II	Single arm: Chemotherapy (S-1 + cisplatin) with trastuzumab	14.6 M	7.4 M	53.3%
	Trastuzumab	first-line [78]	II	Single arm: chemotherapy (CapeOx) with trastuzumab	21.0 M	9.8 M	67%
	Trastuzumab	first-line [79]	II	Single arm: chemotherapy (docetaxel + capecitiabine)	20.9 M	8.1 M	67.8%
	Trastuzumab	first-line [80]	II	Single arm: chemotherapy (S-1 + oxaliplatin) with trastuzumab	18.1 M	8.8 M	70.7%
	Margetuximab	second-line [81]	Ib/II	Margetuximab + pembrolizumab	12.5 M	2.73 M	18.48%
	Trastuzumab	ToGA/first-line [82]	III	Chemotherapy (cisplatin + 5-FU) with or without trastuzumab	13.8 M vs. 11.1 M (*p* = 0.0046)	6.7 M vs. 5.5 M (*p* = 0.0002)	47% vs. 35% (*p* = 0.0017)
	Trastuzumab	HELOISE/first-line [83]	IIIb	Chemotherapy (capecitabine + cisplatin) with standard-dose vs. high-dose trastuzumab	12.5 M vs. 10.6 M (*p* = 0.2401)	5.7 M vs. 5.6 M (*p* = 0.8222)	58.9% vs. 56.9% (*p* = 0.76)
	Trastuzumab/Pertuzumab	JACOB/first-line [84]	III	Trastuzumab + pertuzumab + chemotherapy (capecitabine, cisplatin, or 5-FU) vs. trastuzumab + placebo + chemotherapy	17.5 M vs. 14.2 M (*p* = 0.057)	8.5 M vs. 7.0 M (*p* = 0.0001)	56.7% vs. 48.3% (*p* = 0.026)
ADCs	Trastuzumab deruxtexan	DESTINY-Gastric01/second-line [85]	II	Chemotherapy or trastuzumab deruxtecan	12.5 M vs. 8.4 M (*p* = 0.01)	5.6 M vs. 3.5 M (*p* = N/A)	51% vs. 14% (*p* < 0.001)
	RC48-ADC	second-line [86]	II	Single arm: RC48	7.9 M	4.1 M	18.1%
	Trastuzumab emtansine (T-DM1)	GATSBY/second-line [87]	III	T-DM1 vs. taxane	7.9 M vs. 8.6 M (*p* = 0.86)	2.7 M vs. 2.9 M (*p* = 0.31)	20.6% vs. 19.6% (*p* = 0.846)
TKIs	Dacomitinib	second-line [88]	II	Single arm: Dacomitinib	7.1 M	2.1 M	7.4%
	Lapatinib	first-line [89]	II	Single arm: lapatinib	4.8 M	1.9 M	11%
	Lapatinib	LoGIC/first-line [90]	III	Chemotherapy with or without lapatinib	12.2 M vs. 10.5 M (*p* = 0.91)	6.0 M vs. 5.4 M (*p* = 0.0381)	53% vs. 39% (*p* = 0.0031)
	Lapatinib	TyTAN/second-line [91]	III	Paclitaxel with or without lapatinib	11.0 M vs. 8.9 M (*p* = 0.1044)	5.4 M vs. 4.4 M (*p* = 0.2441)	27% vs. 9% (*p* < 0.001)

**Table 3 cancers-14-05615-t003:** Phase II and III trials of therapies targeting angiogenesis. mOS, median overall survival; mPFS, median progression-free survival; HR, hazard ratio; M, months; ORR, objective response rate; 5-FU, 5-fluorouracil; FOLFOX, folinic acid, fluorouracil, oxaliplatin, oxaliplatin; FOLFIRI, folinic acid, fluorouracil, irinotecan; N/A, not available; TKI, tyrosine kinase inhibitor.

Drug Class	Drug	Trial	Phase	Method	mOS (*p*-Value)	mPFS (*p*-Value)	ORR (*p*-Value)
Monoclonal antibody	Bevacizumab	first-line [107]	II	Single arm: Bevacizumab + docetaxel + 5-FU	16.8 M	6-month PFS rate: 79%	67%
	Bevacizumab	first-line [108]	II	Single arm: Bevacizumab + docetaxel + cisplatin + irinotecan	N/A	N/A	69%
	Bevacizumab	first-line [109]	II	Single arm: Bevacizumab + docetaxel + oxaliplatin	11.1 M	6.6 M	42%
	Bevacizumab	first-line [110]	II	Single arm: Bevacizumab + irinotecan + cisplatin	12.3 M	8.3 M	65%
	Bevacizumab	first-line [111]	II	Single arm: Bevacizumab + capecitabine + oxaliplatin	10.8 M	7.2 M	51.4%
	Bevacizumab	first-line [112]	II	Single arm: Bevacizumab + carboplatin + capecitabine	14.3 M	8.5 M	51%
	Bevacizumab	first-line [113]	Ib/II	Single arm: bevacizumab + docetaxel + capecitabine + cisplatin	13.9 M	7.6 M	54%
	Bevacizumab	first-line [114]	II	Single arm: Bevacizumab + docetaxel + oxaliplatin + capecitabine + trastuzumab	17.9 M	10.8 M	74%
	Bevacizumab	first-line [115]	II	Single arm: Bevacizumab + docetaxel + oxaliplatin + capecitabine	12.0 M	8.3 M	70%
	Bevacizumab	GASTRIC-3/first-line [116]	II	Single arm: oxaliplatin + irinotecan → bevacizumab + docetaxel	11.0 M	7.0 M	51.5%
	Bevacizumab	first-line [117]	II	Single arm: mFOLFOX5 + Bevacizumab	14.7 M	7.8 M	56.4%
	Ramucirumab	first-line [118]	II	Chemotherapy (mFOLFOX6) with or without ramucirumab	11.7 M vs. 11.5 M (*p* = 0.712)	6.4 M vs. 6.7 M (HR = 0.98, *p* = 0.886)	45.2% vs. 46.4% (*p* = 0.830)
	Ramucirumab	RAINSTORM/first-line [119]	II	Chemotherapy (S-1 + oxaliplatin) with or without ramucirumab	N/A	6.34 M vs. 6.74 M (*p* = 0.698)	58% vs. 50% (*p* = 0.402)
	Ramucirumab	REGARD/second-line [120]	II	Single arm: Ramucirumab	8.6 M	6.6 weeks	0%
	Bevacizumab	AVAGAST/first-line [121]	III	Chemotherapy (5-FU + cisplatin + capecitabine) with or without bevacizumab	12.1 M vs. 10.1 M (*p* = 0.1002)	6.7 M vs. 5.3 M (*p* = 0.0037)	46.0% vs. 37.4% (*p* = 0.0315)
	Bevacizumab	AVATAR/first-line [122]	III	Chemotherapy (capecitabine + cisplatin) with or without bevacizumab	10.5 M vs. 11.4 M (*p* = 0.5567)	6.3 M vs. 6.0 M (*p* = 0.4709)	40.7% vs. 33.7% (*p* > 0.05)
	Ramucirumab	RAINFALL/first-line [123]	III	Chemotherapy (cisplatin + 5-FU/capecitabine) with or without ramucirumab	11.2 M vs. 10.7 M (*p* = 0.6757)	5.7 M vs. 5.4 M (*p* = 0.0106)	41% vs. 36% (*p* = 0.17)
	Ramucirumab	REGARD/second-line [124]	III	Ramucirumab vs. placebo	5.2 M vs. 3.8 M (*p* = 0.047)	2.1 M vs. 1.3 M (*p* < 0.0001)	3% vs. 3% (*p* > 0.05)
	Ramucirumab	RAINBOW/second-line [125]	III	Chemotherapy (paclitaxel) with or without ramucirumab	9.6 M vs. 7.4 M (*p* = 0.017)	4.4 M vs. 2.9 M (*p* < 0.0001)	28% vs. 16% (*p* = 0.0001)
TKI	Sorafenib	ECOG5203/first-line [126]	II	Chemotherapy (docetaxel + cisplatin) with or without sorafenib	13.6 M	5.8 M	41%
	Sorafenib	first-line [127]	II	Chemotherapy (capecitabine + cisplatin) with or without sorafenib	11.7 M vs. 10.8 M (*p* = 0.661)	5.6 M vs. 5.3 M (*p* = 0.609)	54% vs. 52% (*p* = 0.826)
	Sorafenib	GEMCAD/second-line [128]	II	Single arm: Chemotherapy (oxaliplatin) with sorafenib	6.5 M	3 M	N/A
	Sorafenib	≥second-line [129]	II	Single arm: sorafenib	9.7 M	3.6 M	3%
	Sunitinib	second-line [130]	II	Single arm: sunitinib	6.8 M	2.3 M	2.6%
	Sunitinib	second-line [131]	II	Chemotherapy (docetaxel) with or without sunitinib	8.0 M vs. 6.6 M (*p* = 0.802)	3.9 M vs. 2.6 M (*p* = 0.206)	41.1% vs. 14.3% (*p* = 0.002)
	Sunitinib	≥second-line [132]	II	Chemotherapy (Na-FOLFIRI) with or without sunitinib	10.4 M vs. 8.9 M (*p* = 0.21)	3.5 M vs. 3.3 M (*p* = 0.66)	20% vs. 29% (*p* = N/A)
	Sunitinib	≥second-line [133]	II	Single arm: sunitinib	5.81 M	1.28 M	3.9%
	Telatinib	TEL0805/first-line [134]	II	Single arm: chemotherapy (capecitabine + cisplatin) with telatinib	N/A	4.7 M	67%
	Orantinib	first-line [135]	II	Chemotherapy (S-1 + cisplatin) with or without oratinib	16.6 M vs. 15.5 M (*p* = 0.213)	6.9 M vs. 7.1 M (*p* = 0.424)	62.2% vs. 56.5% (*p* = 0.671)
	Apatinib	≥third-line [136]	II	Placebo vs. apatinib (850 mg) vs. apatinib (425 mg bid)	2.50 M vs. 4.83 M vs. 4.27 M (*p* < 0.05)	1.40 M vs. 3.67 M vs. 3.20 M (*p* < 0.001)	0% vs. 6.38% vs. 13.04% (*p* = N/A)
	Pazopanib	first-line [137]	II	Single arm: chemotherapy (CapeOx) with pazopanib	10.5 M	6.5 M	62.4%
	Pazopanib	first-line [138]	II	Chemotherapy (5-FU + oxaliplatin) with or without pazopanib	10.1 M vs. 7.0 M (*p* = N/A)	5.1 M vs. 3.9 M (*p* = N/A)	N/A
	Regorafenib	first-line [139]	II	Single arm: chemotherapy (mFOLFOX6) with regorafenib	14.2 M	7.1 M	54%
	Regorafenib	≥second-line [140]	II	Regorafenib vs. placebo	5.8 M vs. 4.5 M (*p* = 0.147)	2.6 M vs. 0.9 M (*p* < 0.001)	N/A
	Fruquintinib	second-line [141]	I/II	Single arm: fruquintinib with paclitaxel	8.5 M	4.0 M	25.9%
	Lenvatinib	≥first-line [142]	II	Single arm: Lenvatinib + pembrolizumab	N/A	6.9 M	69%
	Lenvatinib	≥third-line [143]	II	Single arm: Lenvatinib + pembrolizumab	5.9 M	2.5 M	10%
	Apatinib	≥third-line [144]	III	Apatinib vs. placebo	6.5 M vs. 4.7 M (*p* = 0.0149)	2.6 M vs. 1.8 M (*p* < 0.001)	2.84% vs. 0.00% (*p* = 0.1695)
Recombinant fusion protein	Ziv-aflibercept	first-line [145]	II	Chemotherapy (mFOLFOX6) with or without ziv-aflibercept	14.5 M vs. 18.8 M (*p* = 0.45)	9.7 M vs. 7.4 M (*p* = 0.72)	61.1% vs. 75.0% (*p* = 0.53)

**Table 4 cancers-14-05615-t004:** Phase II and III trials of immune checkpoint inhibitors for gastric cancer. mOS, median overall survival; mPFS, median progression-free survival; HR, hazard ratio; M, months; ORR, objective response rate; 5-FU, 5-fluorouracil; FOLFOX, folinic acid, fluorouracil, oxaliplatin, oxaliplatin; FOLFIRI, folinic acid, fluorouracil, irinotecan; N/A, not available; TKI, tyrosine kinase inhibitor; PD-1, programmed cell death protein-1; PD-L1, programmed cell death ligand 1; CTLA-4, cytotoxic T lymphocyte protein 4; DKK1, Dickkopf-1; CPS, combined positive score.

Drug Class	Drug	Trial	Phase	Method	mOS (*p*-Value)	mPFS (*p*-Value)	ORR (*p*-Value)
Anti-PD-1	Pembrolizumab	KEYNOTE-059/≥second-line [154]	II	Single arm: pembrolizumab	2.0 M	5.6 M	PD-L1-positive tumor: 15.5% PD-L1-negative tumor: 6.4%
	Pembrolizumab	PANTHERA/first-line [155]	Ib/II	Single arm: Chemotherapy (capecitabine + cisplatin) + pembrolizumab + trastuzumab	19.3 M	8.6 M	76.7%
	Pembrolizumab	first-line [156]	II	Single arm: pembrolizumab + trastuzumab	27.2 M	13.0 M	91%
	Pembrolizumab	EPOC1706/≥first-line [142]	II	Single arm:lenvatinib + pembrolizumab	NR	6.9 M	69%
	Pembrolizumab	≥ third-line [143]	II	Single arm: Lenvatinib + pembrolizumab	5.9 M	2.5 M	10%
	Pembrolizumab	second-line [81]	Ib/II	Margetuximab + pembrolizumab	12.5 M	2.73 M	18.48%
	Pembrolizumab	≥second-line [157]	Ib/II	Single arm: DKN-01 + pembrolizumab	DKK1 high: 7.3 M DKK1 low: 4.0 M	DKK1 high: 5.1 M DKK1 low: 1.4 M	DKK1 high: 50% DKK1 low: 0%
	Nivolumab	second-line [158]	Ib/II	Single arm: paclitaxel + nivolumab + ramucirumab	13.1 M	5.1 M	37.2%
	Nivolumab	NivoRAM/second-line [159]	I/II	Single arm: nivolumab + ramucirumab	9.0 M	2.9 M	26.7%
	Camrelizumab	first-line [160]	II	Single arm: CAPOX + camrelizumab → camrelizumab + apatinib	14.9 M	6.8 M	58.3%
	Sintilimab	first-line [161]	II	Single arm: Chemotherapy (CAPOX) with sintilimab	N/A	N/A	85.0%
	Toripalimab	first-line [162]	Ib/II	Toripalimab alone vs. chemotherapy (CAPOX) with toripalimab	4.8 M vs. NR	1.9 M vs. 5.8 M	12.1% vs. 66.7%
	Tislelizumab	first-line [163]	II	Single arm: chemotherapy (CAPOX) + tislelizumab	NR	6.1 M	46.7%
	Pembrolizumab	KEYNOTE-061/second-line [164]	III	Pembrolizumab vs. paclitaxel	9.1 M vs. 8.3 M (*p* = 0.0421)	1.5 M vs. 4.1 M (*p* = N/A)	N/A
	Pembrolizumab	KEYNOTE-062/first-line [165]	IIIs	Pembrolizumab vs. chemotherapy (cisplatin + 5-FU/capecitabine) + pembrolizumab vs. chemotherapy with placebo	CPS ≥ 1: 10.6 M vs. 12.5 M vs. 11.1 M CPS ≥ 10: 17.4 M vs. 12.3 M vs. 10.8 M	CPS ≥ 1: 2.0 M vs. 6.9 M vs. 6.4 M CPS ≥ 10: 2.9 M vs. N/A vs. 6.1 M	CPS ≥ 1: 15% vs. 49% vs. 37% CPS ≥ 10: 25% vs. 53% vs. 38%
	Pembrolizumab	KEYNOTE-811/first-line [166]	III	Trastuzumab + chemotherapy (CAPOX/5-FU + cisplatin) with or without pembrolizumab	N/A	N/A	74.4% vs. 51.9% (*p* = 0. 00006)
	Pembrolizumab	LEAP-005/≥third-line [143]	II	Single arm: Lenvatinib + pembrolizumab	5.9 M	2.5 M	10%
	Nivolumab	ATTRACTION-2/≥second-line [167]	III	Nivolumab vs. placebo	5.3 M vs. 4.1 M (*p* < 0.0001)	1.61 M vs. 1.45 M (*p* < 0.0001)	11.2% vs. 0%
	Nivolumab	CheckMate-649/first-line [168]	III	Nivolumab + chemotherapy (CAPOX or FOLFOX) vs. chemotherapy alone	13.8 M vs. 11.6 M (*p* < 0.0002)	7.7 M vs. 6.9 M (*p* = N/A)	60% vs. 45%
	Nivolumab	CheckMate-577/adjuvant [169]	III	Nivolumab vs. placebo	DFS: 22.4 M vs. 11.0 M (*p* < 0.001)	N/A	N/A
Anti-PD-L1	Avelumab	JAVELIN Gastric 100/first-line [170]	III	Chemotherapy (5-FU + oxaliplatin) with or without avelumab	10.4 M vs. 10.9 M (*p* = 0.1779)	3.2 M vs. 4.4 M (*p* = N/A)	13.3% vs. 14.4% (*p* = N/A)
Anti-CTLA-4	Ipilimumab	first-line [171]	II	Chemotherapy (5-FU + platinum) with or without ipilimumab	12.7 vs. 12.1 (*p* = N/A)	2.7 M vs. 4.9 M (*p* = 0.034)	1.8% vs. 7.0% (*p* = N/A)
Anti-PD-1/CTLA-4	Nivolumab, ipilimumab	CheckMate-032/≥second-line [172]	I/II	Nivolumab (3 mg/kg) vs nivolumab (1 mg/kg) with ipilimumab (3 mg/kg) vs nivolumab (3 mg/kg) with ipilimumab (1 mg/kg)	6.2 M vs. 6.9 M vs. 4.8 M	1.4 M vs. 1.4 M vs. 1.6 M	12% vs. 24% vs. 8%
	Cadonilimab	AK104/first-line [173]	Ib/II	Single arm: Chemotherapy (CAPOX) + cadonilimab	17.41 M	7.10 M	65.9%
Anti-PD-L1/CTLA-4	Durvalumab, tremelimumab	≥second-line [174]	Ib/II	second-line durvalumab with tremelimumab vs. third-line durvalumab with tremelimumab vs. second-line durvalumab alone	9.2 M vs. 10.6 M vs. 3.2 M	1.8 M vs. 1.8 M vs. 1.6 M	11.1% vs. 12.0% vs. 8.3%

**Table 5 cancers-14-05615-t005:** Phase II and III trials of therapies targeting DNA synthesis. mOS, median overall survival; mPFS, median progression-free survival; HR, hazard ratio; M, months; ORR, objective response rate.

Drug	Trial	Phase	Method	mOS (*p*-Value)	mPFS (*p*-Value)	ORR (*p*-Value)
TAS-102 (Trifluridine/tipiracil)	EPOC1201/≥ second-line [183]	II	Single arm: TAS-102 (trifluridine/tipiracil)	8.7 M	2.9 M	3.4%
	≥second-line [179]	II	Single arm: TAS-102 + ramucirumab	6.2 M	4.9 M	N/A
	TAGS/≥ second-line [180]	III	Trifluridine/tipiracil vs. placebo	5.7 M vs. 3.6 M (*p* = 0.00058)	2.0 M vs. 1.8 M (*p* < 0.0001)	4% vs. 2% (*p* = 0.28)
	TAGS/third-line [181]	III	Trifluridine/tipiracil vs. placebo	6.8 M vs. 3.2 M (*p* = 0.0318)	3.1 M vs. 1.9 M (*p* = 0.0004)	N/A
	TAGS/≥ fourth-line [181]	III	Trifluridine/tipiracil vs. placebo	5.2 M vs. 3.7 M (*p* = 0.0192)	1.9 M vs. 1.8 M (*p* < 0.0001)	N/A

**Table 6 cancers-14-05615-t006:** Phase II and III trials of therapies targeting hepatocyte growth factor receptor-1. mOS, median overall survival; mPFS, median progression-free survival; HR, hazard ratio; M, months; ORR, objective response rate; FOLFOX, folinic acid, fluorouracil, oxaliplatin.

Drug Class	Drug	Trial	Phase	Method	mOS (*p*-Value)	mPFS (*p*-Value)	ORR (*p*-Value)
Monoclonal antibody	Rilotumumab	first-line [186]	II	Chemotherapy (epirubicin + cisplatin + capecitabine) with either rilotumumab 15 mg/kg, rilotumumab 7.5 mg/kg, or placebo	N/A	5.1 M vs. 6.8 M 4.2 M	N/A
	Rilotumumab	MEGA/first-line [71]	II	Chemotherapy with or without either panitumumab or rilotumumab	8.3 M (P) vs. 11.5 M (R) vs. 13.1 M (C)	4-month PFS rate: 57% (P) vs. 61% (R) vs. 71% (C)	43% (P) vs. 49% (R) vs. 52% (C)
	Onartuzumab	YO28252/first-line [187]	II	Chemotherapy (mFOLFOX6) with or without onartuzumab	10.6 M vs. 11.3 M (*p* = 0.83)	5.95 M vs. 6.80 M (*p* = 0.45)	60.5% vs. 57.1%
	Emibetuzumab	≥third-line [188]	II	Single arm: emibetuzumab	3.9 M	1.9 M	N/A
	Emibetuzumab	≥first-line [189]	Ib/II	Single arm: ramuricumab + emibetuzumab	N/A	1.6 M	6%
	Rilotumumab	RILOMET-1/first-line [190]	III	Chemotherapy (epirubicin + cisplatin + capecitabine) with or without rilotumumab	8.8 M vs. 10.7 M (*p* = 0.003)	5.6 M vs. 6.0 M (*p* = 0.016)	29.8% vs. 44.6% (*p* = 0.0005)
	Onartuzumab	METGastric/first-line [191]	III	Chemotherapy (mFOLFOX6) with or without onartuzumab	11.0 M vs. 11.3 M (*p* = 0.24)	6.7 M vs. 6.8 M (*p* = 0.43)	46.1% vs. 40.6% (*p* = 0.25)
TKIs	Foretinib	≥first-line [192]	II	Single arm: foretinib intermittent dosing vs. daily dosing	Intermittent: 7.4 M Daily: 4.3 M	Intermittent: 1.6 M Daily: 1.8 M	0%
	Tivantinib	first-line [193]	II	Single arm: chemotherapy (FOLFOX) with tinvantinib	9.6 M	6.1 M	38%

**Table 7 cancers-14-05615-t007:** Phase II and III trials of therapies targeting fibroblast growth factor receptor. mOS, median overall survival; mPFS, median progression-free survival; M, months; ORR, objective response rate; FOLFOX, folinic acid, fluorouracil, oxaliplatin.

Drug Class	Drug	Trial	Phase	Method	mOS (*p*-Value)	mPFS (*p*-Value)	ORR (*p*-Value)
Monoclonal antibody	Bemarituzumab	FIGHT/first-line [195]	II	Chemotherapy (mFOLFOX6) with or without bemarituzumab	NR vs. 12.9 M (*p* = 0.03)	9.5 M vs. 7.4 M (*p* = 0.07)	53% vs. 40%
TKIs	AZD4547	SHINE/second-line [196]	II	AZD4547 vs. chemotherapy (paclitaxel)	5.5 M vs. 6.6 M (*p* = 0.82)	1.8 M vs. 3.5 M (*p* = 0.96)	2.6% vs. 23.3% (*p* = 0.99)

**Table 8 cancers-14-05615-t008:** Phase II and III trials of therapies targeting poly (ADP-ribose) polymerase. mOS, median overall survival; mPFS, median progression-free survival; HR, hazard ratio; M, months; ORR, objective response rate.

Drug	Trial	Phase	Method	mOS (*p*-Value)	mPFS (*p*-Value)	ORR (*p*-Value)
Pamiparib	PARALLEL 303/>first-line [200]	II	Pamiparib vs. placebo	10.2 M vs. 12.0 M (*p* = N/A)	3.7 M vs. 2.1 M (*p* = 0.14)	7.7% vs. 6.3%
Olaparib	>first-line [198]	II	Chemotherapy (paclitaxel) with or without olaparib	13.1 M vs. 8.3 M (*p* = 0.005)	3.91 M vs. 3.55 M (*p* = 0.131)	26.4% vs. 19.1% (*p* = 0.162)
	GOLD/>first-line [199]	III	Chemotherapy (paclitaxel) with or without olaparib	8.8 M vs. 6.9 M (*p* = 0.026)	3.7 M vs. 3.2 M (*p* = 0.064)	24% vs. 28% (*p* = 0.055)

**Table 9 cancers-14-05615-t009:** Phase II and III trials of therapies targeting matrix metalloproteinase-9. mOS, median overall survival; mPFS, median progression-free survival; HR, hazard ratio; M, months; ORR, objective response rate.

Drug Class	Drug	Trial	Phase	Method	mOS (*p*-Value)	mPFS (*p*-Value)	ORR (*p*-Value)
ICI + monoclonal antibody	Andecaliximab	≥first-line [201]	II	Andecaliximab + nivolumumab vs. nivolumumab alone	7.1 M vs. 5.9 M (*p* = 0.23)	N/A (*p* = 0.10)	9.7% vs. 6.9% (*p* = 0.8)
Monoclonal antibody	Andecaliximab	GAMMA-1/first-line [202]	III	Chemotherapy (mFOLFOX6) with or without andecaliximab	12.5 M vs. 11.8 M (*p* = 0.56)	7.5 M vs. 7.1 M (*p* = 0.10)	50.5% vs. 41.1%

**Table 10 cancers-14-05615-t010:** Phase II and III trials of therapies targeting mammalian target of rapamycin mOS, median overall survival; mPFS, median progression-free survival; HR, hazard ratio; M, months; ORR, objective response rate.

Drug	Trial	Phase	Method	mOS (*p*-Value)	mPFS (*p*-Value)	ORR (*p*-Value)
Everolimus	>first-line [204]	II	Single arm: everolimus	10.1 M	2.7 M	N/A
Everolimus	GRANITE-1/>first-line [203]	III	Everolimus vs. placebo	5.4 M vs. 4.3 M (*p* = 0.124)	1.7 M vs. 1.4 M (*p* = N/A)	4.5% vs. 2.1% (*p* = N/A)

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
