# Peer review of "Recent Trends and Advancements in the Diagnosis and Management of Gastric Cancer"

_cancers, 2022, doi:10.3390/cancers14225615_

Round 1

Reviewer 1 Report

This paper shows a very good and comprehensive writing on the topic of recent trend and advancement in the diagnosis and management of gastric cancer. A few minor corrections as below:

Line 43: the word cagA and H. pylori should be written in italic.

Line 142: Mistake in numbering of the subtopic. It should be 2.1.4

Line 180: Add figure number before the title and cite the figure in the text.

Line 186: I think the sentence in not complete. Please revise.

All tables should be numbered accordingly and write the title for each table. Also citing of the tables in respective text is required.

Reviewer 2 Report

Haque and collaborators provided an overview of the GC diagnostic markers currently available in clinical practice and clinical trials aimed at expanding the available therapies. Although the subject matter is of great interest, some points need substantial revision.

- Major revisions

1-The authors focused their attention on diagnostic markers whose relevance in clinical practice is based on non-invasive detection methods. In this area, the authors are to discuss other markers that have not been referred to.  The role of AFP, CA125 in neovascularization or peritoneal dissemination as well as the recent finding about exosomes as novel diagnostic biomarkers need to be discussed.

2- In section 3, focusing on treatments, GC-associated heterogeneity should be further discussed with reference to both clinical and molecular heterogeneity. Therefore, the new classification of GC patients can provide approaches and therapeutics based on the genome and clinical evidence. These aspects should also be discussed in the conclusions.

Minor revisions

- Affiliations 5 and 6 do not correspond to any author

Reviewer 3 Report

This paper is a review about gastric cancer, mainly focused on the diagnosis and treatment except endoscopic and surgical treatment. This paper well summarized
about molecular targeted therapy with a bunch of trial chemotherapies.

Round 2

Reviewer 2 Report

The authors revised the manuscript fully responding to the criticism I raised. I thus consider the manuscript suitable for publication in the present form.

Author Response

Thank you very much for your comments and efforts you put in for the revision, we really appreciate it.